# Fatigue Strength of Structural Steel-Welded Connections with Arc-Sprayed Aluminum Coatings and Corrosion Behavior of the Corresponding Coatings in Sea Water

Andreas Gericke [1,*], Michél Hauer [1] 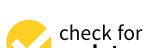, Benjamin Ripsch [1], Michael Irmer [2], Jonas Nehlsen [2] and Knuth-Michael Henkel [3]

1 Thermal Joining Engineering, Fraunhofer Institute for Large Structures in Production Engineering IGP, 18059 Rostock, Germany
2 Coating, Weathering and Corrosion Protection, Fraunhofer Institute for Large Structures in Production Engineering IGP, 18059 Rostock, Germany
3 Faculty of Mechanical Engineering and Marine Technologies, Joining Technology, University of Rostock, 18059 Rostock, Germany
* Correspondence: andreas.gericke@igp.fraunhofer.de; Tel.: +49-381-49682-37

**Abstract:** The influence of thermally sprayed aluminum coatings (Al99%; arc spraying) on the fatigue strength of gas metal arc welded (GMAW) non-alloyed structural steel specimens with respect to foundations for offshore wind turbines was investigated. Additionally, the corrosion protection effect of such coatings for water conditions similar to the Baltic Sea was determined. Wöhler tests were carried out on test specimens with different weld details in the as-welded condition as well as in the thermal spray coat under the consideration of different kinds of surface preparation (blast cleaning with corundum and grit). Substrate and coating were characterized by scanning electron microscopy and the influence on the residual stress states was determined. Corrosion rate monitoring via LPR measurements was carried out as well as the monitoring of the galvanic current between coated and uncoated steel to characterize the coatings' sacrificial capability for minor defects. Fatigue strength was significantly increased through thermal spraying, especially for test specimens with welded transverse stiffeners ($\Delta\sigma_{c,var}$ = 127 MPa after coating compared to $\Delta\sigma_{c,var}$ = 89 MPa as welded). With a characteristic value of the stress range of $\Delta\sigma_{c,var}$ = 153 MPa, the welded butt joint specimens already exhibited a high fatigue strength in the as-welded condition. The corrosion studies demonstrated that thermally sprayed Al99% coatings have a high resistance to corrosion in seawater environments and are suitable as planar sacrificial anodes sufficiently polarizing bare steel below 0.8 V. The combination of fatigue strength improvement and corrosion protection makes the thermally sprayed Al coatings promising for design and operation of e.g., offshore structures.

**Keywords:** fatigue; corrosion; welding; thermal spraying; Al99; offshore; structural steel; blast cleaning; LPR; GMAW

## 1. Introduction

For offshore wind energy plants, achieving the longest possible service life of foundation and tower structure is decisive for their economic efficiency. Due to challenging mechanical and corrosive conditions, structural steel foundations under water are at a high risk of degradation. Cyclic loading from waves, wind and plant operation can lead to $10^8$ load cycles or more within the aspired 25 years of service. The design service life of towers and tower foundations is mainly determined by the fatigue strength of their welded connections. Apart from the apparent butt-welded connections, the attachment of support elements (non-load carrying welded attachments; detail category 80 acc. to EN 199319 [1]) limits the fatigue strength of the structure and leads to the use of plate thicknesses of more than 130 mm [2].

The fatigue strength of welded components is mainly determined by the geometrical and technological notch effect of the welds. The higher the quality level of the welds, the less distinct the notch effect and the better the fatigue strength of the welded detail. Other factors, such as the base material strength or welding process, are of minor importance for the fatigue strength of welded components [3]. Offshore wind turbine towers and tower foundations are manufactured at onshore facilities with high standards regarding the quality level of execution and quality assurance, so limitations in fatigue strength can only be raised by applying post-weld treatment methods.

Different kinds of post-weld treatment methods are frequently used for improving the fatigue strength of welded connections. Principally, post-weld treatment methods rely on reducing the notch effect of the weld by the reduction of stress concentration at the weld toe and on the introduction of beneficial compressive residual stresses [4]. In addition to the more commonly known and regulated burr grinding, TIG dressing, and HFMI processes, blast cleaning can lead to significant increases in fatigue strength of structural steel as well. Before coating, the welded components of wind turbine towers are blast cleaned for surface preparation, so the process can be used as an economic post-weld treatment method that does not require additional manufacturing steps [5–7].

Generally, post-weld treatment methods are also available for offshore foundations. However, because of the additional manufacturing effort and the comparatively small benefit for design due to regulation constraints, they are rarely implemented in production. Also, wind turbine foundations are rarely coated, so the corrosive effect of seawater must be expected to diminish the positive effect of potential post-weld treatment methods. Enhanced benefits from post-weld treatment methods in structural components located under water can be achieved by considering the coating of the corresponding areas.

Until recently, only active (ICCP) and passive (sacrificial anodes) corrosion protection systems have been used in the underwater area of wind turbine foundations. A new and increasingly used method for corrosion protection is the aluminum coating of the entire foundation structure by means of thermal spraying, see Figure 1. Here, the entire structure exposed to seawater is preserved by means of arc spraying with a 99% aluminum coating (Al 99%) as well as an organic sealer [8].

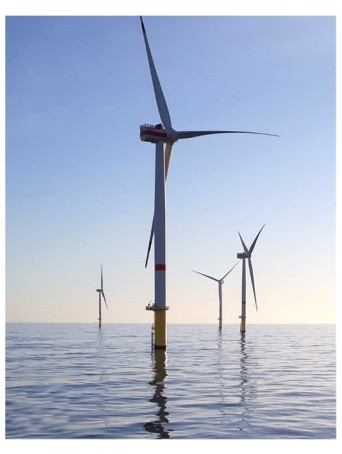

(**a**)

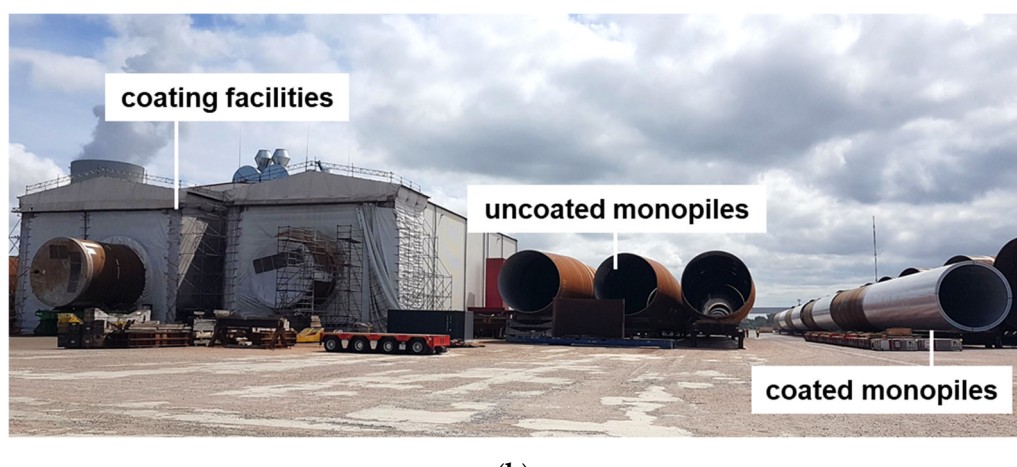

(**b**)

**Figure 1.** (**a**) Offshore wind turbines; (**b**) Facility for thermal spraying of offshore foundations as well as uncoated and coated monopiles.

The arc-spray process is a simple and cost-effective thermal spray technology, which is determined by a few key factors [9,10]. It is well established for the corrosion protection of large structures and on-site repairs [11,12].

The effect of thermal spraying on fatigue strength was investigated in different contexts. Depending on the substrate material, surface preparation, coating process, coating system, and a potential corrosive medium, thermal spraying can lead to the improvement as well

as the deterioration of the substrates' fatigue strength characteristics. The introduction of tensile residual stresses as well as the increase of the component roughness were identified as significant variables influencing the fatigue strength of the coated component [13–18].

Arc voltage and current as well as the gas flow, pressure, and type mainly affect the resulting coating quality [10,19]. In addition, residual stresses inside the coating are heavily influenced by the aforementioned parameters and the spray kinematics [19,20]. In the literature, alternating quenching stresses (generating from particles hindered in shrinking by the substrate as soon as they hit the surface) are reported to dominate the composition of residual stresses in arc spraying, while thermal mismatch stresses (caused by different coefficients of thermal expansion for substrate and coating) also play a major role [19–21]. However, this might be altered by phase changes, e.g., due to heat treatment [21,22]. Concerning the superposition of residual stresses and external stresses or an excess of a certain coating thickness, further studies have revealed pronounced coating delamination and thus significantly reduced service lives [21]. Both findings were confirmed by our own investigations [20]. Furthermore, our investigations on cavitation erosion resistance of thermally sprayed coatings show correlations between the spray parameters, the kinematics, and the gases used as well as the residual stresses of the coatings [22]. Coatings made of fatigue-resistant materials were found to be particularly durable in regard to cavitation erosion. However, the substrate used was pure bulk material instead of welded detail [22]. Consequently, it can be assumed that the longevity and residual stress state of the coating has a significant influence on the fatigue strength of the welded and subsequently blast-cleaned and thermally sprayed connections.

The influence of thermally sprayed Al—coatings on welded steel structures subjected to cyclic loading has not been investigated to date and could be beneficial in terms of service life and manufacturing costs of offshore foundations.

The sacrificial behavior concerning the cathodic protection of exposed steel as well as the self-corrosion rate of arc-sprayed aluminum coatings have been investigated thoroughly [23,24]. However, investigations on the corrosion protection performance as well as on the sacrificial behavior of thermal-spray aluminum coatings in sea water with low salinity via quantitative methods are absent from the literature. Galvanic testing of a combination of thermally coated and uncoated steel can provide insight into the sacrificial behavior of thermal-spray aluminum coatings [25]. Regarding large subsea structures this particular characteristic of thermal-spray aluminum coatings needs to be studied as not only their barrier properties, but also the polarizing effect on mild steel, are key factors to their corrosion protection effect.

Hence, first-time investigations on the fatigue resistance of basic weld details coated with thermally sprayed aluminum under consideration of different surface preparation methods are summarized in this paper. Moreover, investigations regarding the corrosion protection performance and sacrificial behavior of corresponding coatings are presented.

## 2. Materials and Methods

Based on welded connections typical for steel towers and tower foundations, arc-welded butt joint specimens, as well as specimens with transverse stiffeners, were manufactured from plates of non-alloy structural steel EN 10025-2-S355J2+N [26] with a sheet thickness of t = 8 mm. A S5 SpeedPulse XT welding unit (Lorch Schweißtechnik GmbH) was used for welding. A welding tractor was used for guiding the welding torch to ensure reproducible welding results, a constant heat input, and the constant quality of the welds. After welding, the plates were cut into individual specimens on a band saw. Sharp edges were removed by breaking them manually using a file. Specimen geometries as well as the concomitant welding symbols for the GMAW welds are shown in Figure 2.

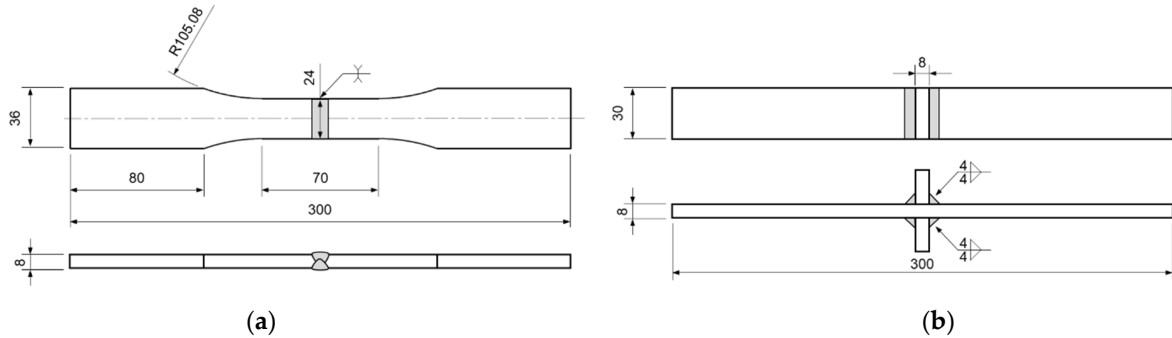

**Figure 2.** Specimen geometry and welding symbols acc. to ISO 2553 [27] (**a**) Butt joint specimen; (**b**) Specimen with transverse stiffeners.

Prior to thermal spraying, the surfaces of the specimens were blast cleaned by using corundum and grit, respectively. Blast cleaning with corundum was performed at the facilities of Linde AG. Grit blasting was carried out at a manufacturer of steel pipes for wind energy turbine towers using blasting parameters typical for surface preparation of the tower segments before coating. To determine the blasting intensity of the blasting processes, Almen intensity tests according to SAE J433 [28] were carried out at each blasting site using a sentenso Almen gage model TSP-3 and Almen test strips Type A (Grade 2) according to SAE J442 [29]. Parameters, consumables, and quality requirements for welding and blasting are shown in Table 1 (heat input calculated with a coefficient of thermal efficiency of k = 0.8 for GMAW processes according to [30,31]).

**Table 1.** Parameters, consumables and quality requirements for welding and blasting.

| Welding Parameter | Value | Blasting Parameters | Value |
|---|---|---|---|
| Arc Voltage | 27 V | Almen Intensity Steel Grit | 628 µmA (T = 2.5 s) |
| Welding Current | 300 A | Almen Intensity Corundum | 467 µmA (T = 18.2 s) |
| Wire feed rate | 10 m/min | **Blasting Consumables** | **Type & Particle Size** |
| Welding speed | 35 cm/min | Steel grit | ISO 11124 M/HCS/G100 [32] |
| Heat input | 1.1 kJ/mm | | nominal particle size 1.0 mm |
| Shielding gas flow rate | 12 L/min | Corundum | F24 (FEPA 42-1 [33]) |
| Electrode stickout | 18–22 mm | | 600–850 µm |
| Welding wire diameter | 1.2 mm | **Blasting Quality Criteria** | **Requirement** |
| Torch angle | 10° | Surface preparation grade | Sa3 (ISO 8501-01 [34]) |
| Travel direction | Push welding | Coverage | 100% (SAE J2277 [35]) |
| **Welding Consumables** | **Designation** | **Welding Quality Criteria** | **Requirement** |
| Welding wire | ISO 17632-A—T 46 6 M M21 1 [36] | General Quality Level | B (ISO 5817 [37]) |
| Shielding gas | ISO 14175—M21—ArC—18 [38] | | |

Thermal spraying of the specimens was carried out at the facilities of Linde AG utilizing the arc spray process. For the spray experiments, a power source Sparc 400 equipped with a Shark 400 RE gun on a robot was used (both GTV Verschleißschutz GmbH, Luckenbach, Germany). The spray parameters for the experiments were kept constant and can be found in Table 2. A meander-shaped type of spray pattern was employed. The specimens were coated on all sides. The two wires corresponded to type Al 99% (Metco Aluminium, Oerlikon Metco GmbH, Kelsterbach, Germany; Ø 1.6 mm).

**Table 2.** Thermal spray parameters.

| Gas | Flow Rate in m³/h | $p_{Gas}$ in Bar | Robot Speed in m/min | Robot Offset in mm | Stand-Off Distance in mm | Number of Passes | Voltage in V | Current in A | Wire Feed Rate in m/min |
|---|---|---|---|---|---|---|---|---|---|
| Air | 93.0 | 3.5 | 40 | 9 | 100 | 2 | 28 | 107 | 6.2 |

Micro-sections of the specimens were prepared by cold mounting (two-phase system: liquid hardener and powder resin), gradual grinding, and subsequent polishing up to 3 μm polishing suspension, which was finished by OP-S polishing. Moreover, Nital etching (3%) was applied for enhancing the contrast.

Micrographs of the micro-sections and fracture surfaces were taken using an optical microscope (OM) Leica DM6000M (Leica Microsystems GmbH, Wetzlar, Germany) and the software DHS tool (dhs GmbH, Greifenstein-Beilstein, Germany). Moreover, representative analyses regarding the morphology were carried out using a scanning electron microscope (SEM) JEOL JSM-IT100 (JEOL Germany GmbH, Freising, Germany; with an acceleration voltage 10 kV, a backscatter detector, and a low vacuum mode). Further investigations were carried out on the microstructure of the substrate materials regarding the impact of blast cleaning and thermal spraying. Additionally, the fracture surfaces were analyzed particularly focusing on the origin of the fatigue cracks. Both were investigated using the same equipment as described earlier.

The coating thickness was determined quantitatively in the OM micrographs, recording 3 times the 7 measured values, while eliminating maxima and minima values for each. In addition, the number of coating defects (i.e., porosity, oxidation, and cracks) was quantified at 3 random areas of the SEM images. For that purpose, the software ImageJ (National Institutes of Health, USA; in the region of interest using Despeckle filter, normalization, and finally the Trainable Weka Segmentation tool) was applied.

Residual stress measurements were carried out for specimen characterization prior to fatigue testing. The measurements were performed by the hole drilling method combined with electronic speckle pattern interferometry (ESPI) using the PRISM system and the PrismS software (both Stresstech GmbH, Rennerod, Germany) with three measurements for each specimen. The coated specimens and the reference material (without grit blasting or thermal spraying) were tested after the application of a developer spray (typically used in the non-destructive penetration test) to reduce surface reflections and, in the case of the coated samples, in the as-sprayed state. The holes were processed in varying steps using high-speed steel end mills (2 fluted, TiN coated, and Ø 0.8 mm) up to a final profile depth of 400 μm. The inner integration radius was twice the hole diameter, while the outer integration radius corresponded to four times the hole diameter. The Poisson ratio used for calculation was 0.334, while the Young's modulus was assumed to be 71 GPA. Both values correspond to bulk aluminum, which should be kept in mind regarding the evaluation. A Tikhonov regularization factor of 0.01 was applied in stress calculation.

Fatigue tests were performed under axial loading and generally according to the ISO/TR 14345 [39]. A resonance testing machine, the Power Swing 100 kN from SincoTec Test Systems GmbH, was used for carrying out the tests. The testing frequency was f = 60 Hz for the butt-welded specimens and f = 110 Hz for the specimens with transverse attachments. Runouts were defined for $5 \times 10^6$ cycles. Specimen failures aimed for an even distribution over the target live range of $5 \times 10^4 < N < 5 \times 10^6$ cycles. A complete crack of the specimens or a visually detectable crack close to being a complete crack were defined as failure criteria. Specimens were loaded by constant amplitude loading with a stress ratio of R = 0.5. The limit to the maximum stress applied was the base material yield strength determined via the tensile tests, which led to a maximum stress range of $\Delta\sigma_{max}$ = 200 MPa. Statistical evaluation of the fatigue test results was carried out according to the background document [40] to the Eurocode 3 Part 1–9 [1]. The runouts as well as specimens with cracks in the base metal area were excluded from the statistical evaluation.

The corrosion protection capability of the Al99% coating was tested in a close-to-real environment. For this purpose, a test stand consisting of 5 Intermediate Bulk Containers (IBC) was set up indoors at a constant temperature of 21 ± 1 °C. Each of the IBCs holds 600 L water, while the one where the testing occurred was also partly filled with sediment. The IBCs are interconnected and a circulative flow of 600 L/h is generated by a submersible pump. Water and sediment were recovered from the Baltic sea at an area 35 km northeast

of the island Rügen at 40 m depth. Table 3 provides an overview of the water composition analyzed by ion chromatography and the additional water properties.

**Table 3.** Seawater composition and properties.

| Component | Sodium | Potassium | Calcium | Magnesium | Chloride | Bromide | Sulphate |
|---|---|---|---|---|---|---|---|
| Concentration in mg/L | 2906.9 | 99.6 | 227.4 | 480.6 | 6518.7 | 47.8 | 800.1 |
| **Property** | **Salinity** | **pH** | **$O_2$** | **Temperature** | | | |
| Value | 0.81 g/L | $7.5 \pm 0.1$ | $7 \pm 0.3$ mg/L | $21 \pm 1\,°C$ | | | |

The coated test specimens are made of S355J2+N Steel with a size of 175 mm × 50 mm × 5 mm. Prior to arc spraying the steel was blast cleaned to Sa 3 to a roughness between 60 μm and 100 μm according to ISO 8501-1 [34]. The coating thickness ranged between 400 μm and 600 μm due to manual application. The bare steel coupons that simulate damage to the coating were sized at 17 mm × 13 mm × 10 mm. All test specimens were provided with a threaded bore with a diameter of 4 mm on the topside for electrical connection. The electrical connection as shown in Figure 3 was established by the standard banana plug measurement cables cut to length and tightened to the coupons. Insulation from the surrounding electrolyte was guaranteed by a silan-modified polymer sealant; with the sealant covering the whole topside of the bare steel coupons and their remaining surface area representing 5% of the surface area of the coated specimens.

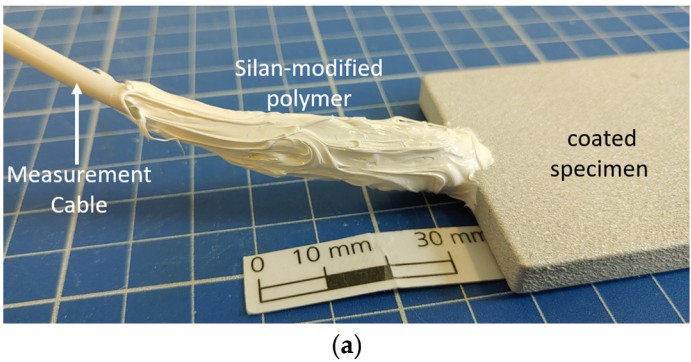 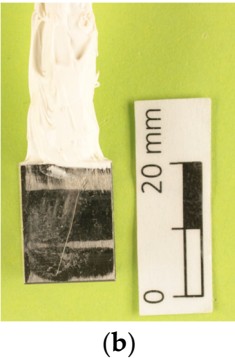

(**a**)        (**b**)

**Figure 3.** (**a**) The electrical connection of a thermal spray-coated specimen with insulation by silan-modified polymer sealant; (**b**) A bare steel specimen with the electrical connection.

Three different specimen configurations were tested simultaneously. Four thermal spray-coated S355J2+N specimens as well as four bare steel specimens were exposed as stand-alone for reference purposes. As a third configuration the thermal spray-coated specimens were electrically coupled to bare steel specimens through a specifically developed Arduino-based connection box in between the measurements and through the potentionstat while measuring. This way an electrical connection between the coated specimens and the bare steel was always provided with the protection current being quantifiable on a regular basis.

Prior to immersion, all specimens were degreased using isopropanol and deionized water. As shown in Figure 4 all configurations were positioned in one IBC around a single Ag/AgCl reference electrode (RE) which was replaced daily. The counter electrode (CE) made from graphite with the dimensions 400 mm × 400 mm × 30 mm was placed on the side of the IBC. The specimens were hung by their cables with a fixed distance of 5 cm (minimum) between the galvanic couples.

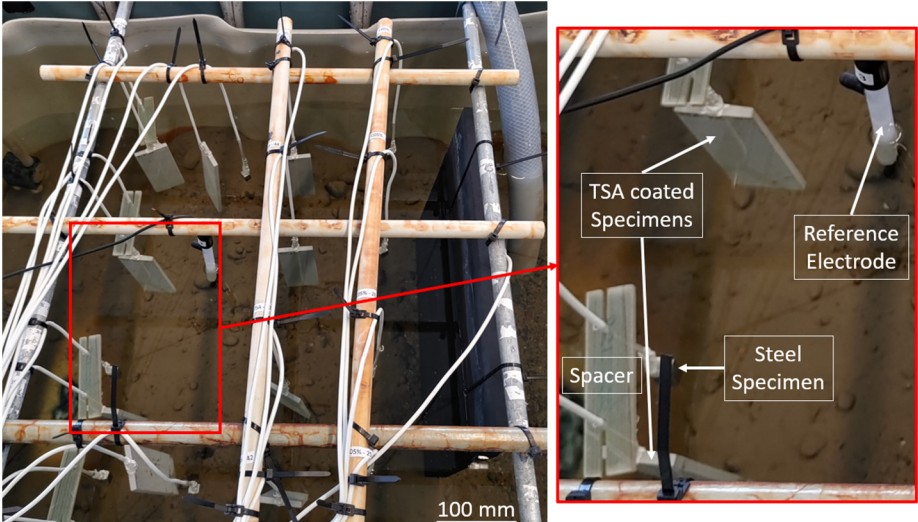

**Figure 4.** The test setup in IBC with the graphite counter electrode to the right, the reference electrode in the middle, and the specimens arranged around it. The galvanic couples are on the left side. The thermal spray-coated specimens are in the middle of the picture and the bare steel specimens are placed to their right.

The potentiostat used was a Metrohm Autolab PGSTAT 302 N equipped with 3 multiplexers for the serial measurement of 12 test specimens or specimen couples. The test duration for the corrosion test was 30 days. At the beginning and end of the test the potentiodynamic polarization curves were recorded using additional specimens. These specimens were polarized down to $-1.2$ V vs. RE-potential before scanning up to $-0.5$ V vs. RE-potential using a scan rate of 0.5 mV/s.

The open circuit potential (OCP) of all specimen configurations and the current between the galvanic couples were recorded every 4 h. At the beginning of each measurement the potentiostat established an electrical measurement circuit parallel to the connection box. After 10 s the wiring through the Arduino-based box was cut for the recording duration (see Figure 2). The values of every channel were tracked for 120 s before reconnecting through the arduino-box and disconnecting the potentiostat.

Once per week and twice during the first week LPR measurements were performed. After 120 s of OCP scanning time the specimens and specimen couples were polarized at $\pm 10$ mV around OCP with a scan rate of 0.1 mV/s. The results from the LPR measurements and the Tafel constants ba and bc gathered from the polarization curves were used to calculate the corrosion rates using the Stern–Geary [41] equation. For the calculation of the corrosion rates of the galvanic couples the equivalent weight of aluminum was used due to the surface area ratio between the coated specimens and the bare steel.

After the exposure period the images in a wet and dried state, the SEM images (same equipment as above), and the energy dispersive x-ray spectrometry (EDX) maps (JEOL Dry SD25 detector (JEOL Germany GmbH, Freising, Germany; acceleration voltage 10 kV)) of selected specimens were acquired. The deposits formed on the coating were analyzed by IR-spectroscopy (FTIR-Spectroscope ALPHA (Bruker Optik GmbH, Leipzig, Germany).

For the base material characterization, the base material yield strength was determined via tensile tests according to ISO 6892-1 [42] and the chemical composition of the base material was verified by optical emission spectrometry (OES) using a Spectromaxx (Spectro Analytical Instruments GmbH, Kleve, Germany).

## 3. Results

### 3.1. Materials Characterization

#### 3.1.1. Base Metal Properties

Tensile tests according to ISO 6892-1 [42] resulted in a medium upper yield strength of $R_{eH}$ = 403 MPa (s = 7 MPa; *n* = 4). The OES analysis results and content limits according to EN 10025-2 [13] are shown in Table 4.

**Table 4.** The OES analysis results and content limits according to EN 10025-2 [13].

| Steel | Content in % | C | Si | Mn | P | S | N | Cu | Ni | Cr | Mo |
|---|---|---|---|---|---|---|---|---|---|---|---|
| S355J2+N acc. to [43] | Max. content acc. to [13] | 0.20 | 0.55 | 1.60 | 0.025 | 0.025 | - | 0.55 | 0.42 | 0.29 | 0.11 |
| | $\bar{x}$ (*n* = 5) | 0.159 | 0.197 | 0.72 | 0.0085 | 0.0042 | 0.022 | 0.021 | 0.042 | 0.056 | 0.012 |
| | $s_x$ | 0.0083 | 0.0038 | 0.0040 | 0.0002 | 0.0001 | 0.0025 | 0.0001 | 0.0021 | 0.0009 | 0.0002 |

#### 3.1.2. Weld Seam Quality

Figure 5 shows the macro sections of a butt-welded specimen and a specimen with transverse stiffeners (both blast cleaned and thermally sprayed). Quality level B according to ISO 5817 [37] was met for the imperfections of the welded joints in the specimens and verified via visual testing.

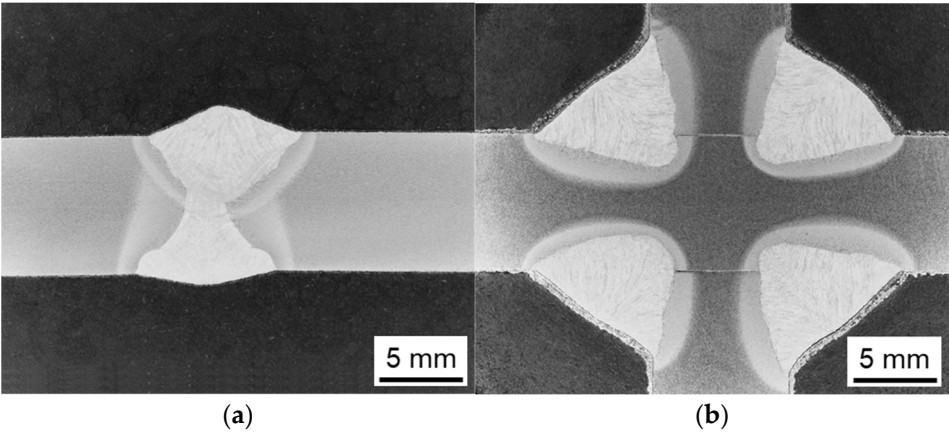

(a)        (b)

**Figure 5.** Macro-sections of thermally sprayed specimens, etched with Nital (5%), (**a**) butt-weld; (**b**) fillet weld.

#### 3.1.3. Surface and Coating Properties

Regarding surface preparation, grit blasting shows a stronger impact on the near surface microstructure of the steel substrate compared to blasting with corundum. A more severe plastic deformation and a visible compression of the microstructure of the surface material layer can be seen in the micro-section of the grit-blasted specimen compared to the specimen blasted with corundum, see Figure 6.

This stronger deformation is also reflected in the subsequent coating analyses. First, it is obvious that the grit-blasted samples exhibit higher coating thicknesses, which are also constant over a larger area in the section, see Figure 7. Apart from this, a typical lamellar morphology with good bonding to the substrate can be observed. Both types of surface preparation share the non-uniform distribution of cluster-like coating defects, which are marked in Figure 7. Yet, these variations are apparently more frequent for the specimens blasted with corundum. In contrast, layered oxides typical for arc spraying are hardly visible. However, coating breakouts can be detected in part for both surface preparations as a result of the polishing process, although they are more pronounced for the specimens blasted with corundum.

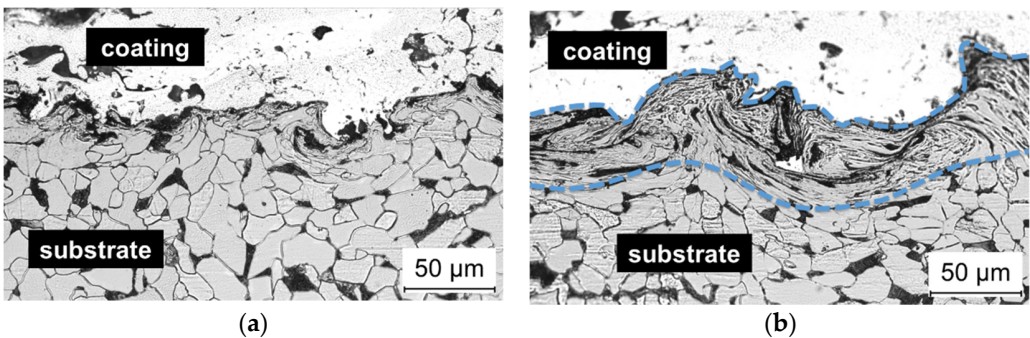

**Figure 6.** The impact of different surface preparation methods on the base metal microstructure; micro-sections etched with Nital (3%), (**a**) substrate blasted with corundum; (**b**) substrate blasted with steel grit (deformed microstructure bordered by the dashed line).

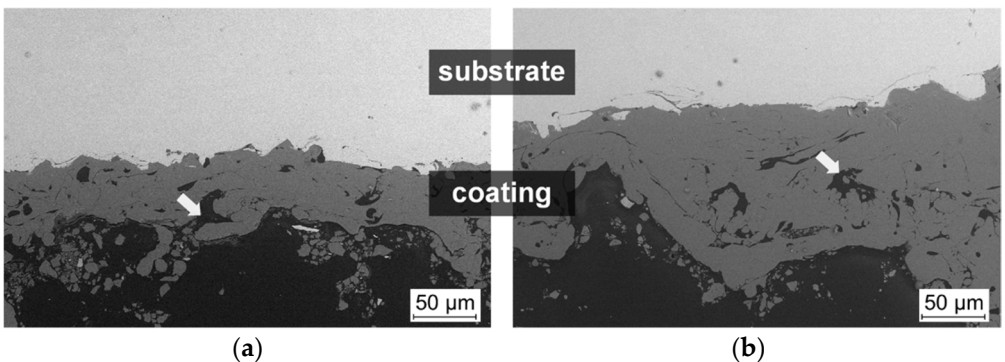

**Figure 7.** Representative SEM images showing the substrate on top and the coating below (specimen bottom) for the samples blasted with (**a**) corundum; (**b**) grit.

The observations on the coating microstructure can further be confirmed quantitatively, see Table 5. For example, the grit-blasted specimens had higher coating thickness than the corundum-blasted specimens, with similar percent standard deviations for both surface preparations.

**Table 5.** The quantitative coating analysis showing thicknesses and defects.

| Blasting Procedure | Coating Thickness in μm | Amount of Defects (Porosity, Cracks, Oxides) in % |
|---|---|---|
| Corundum | 78 ± 20 | 4.7 ± 2.3 |
| Grit | 97 ± 26 | 6.2 ± 0.5 |

Likewise, according to the representative analyses, a uniform coating structure is recognizable for specimens blasted with grit. Although the absolute number of defects is somewhat higher, the higher standard deviation of the corundum blasted samples indicates an increased inhomogeneity.

### 3.1.4. Residual Stress Measurements

The residual stress measurements show a high amount of compressive residual stresses in the blast-cleaned steel substrate (Figure 8). In the as-welded condition, the untreated specimen shows significant tensile residual stresses in the area of the weld transition. The measurements were performed at a distance of 5 mm from the weld toe in the base material. The compressive stresses in the grit-blasted specimens reach deeper underneath the specimen surface and gain a higher compressive stress value of up to −200 MPa compared to the specimens blasted with corundum, with a compressive stress maximum of about −100 MPa, see Figure 8. At a drilling depth within the range of a coating thickness of

approximately 0.05 to 0.1 μm, a transition from tensile stresses to compressive stresses can be seen in the residual stress depth profiles obtained. Tensile residual stresses in the coating of the specimens are higher in the specimens blasted with steel grit and reach levels of up to 200 MPa, compared to less than a 100 MPa tensile stress in the coating of the specimen blasted with corundum. At a depth of 0.35 mm, nearly no compressive stresses could be measured in the specimen blasted with corundum, whereas the grit-blasted specimen still showed compressive stresses of more than $-50$ MPa at the same drilling depth.

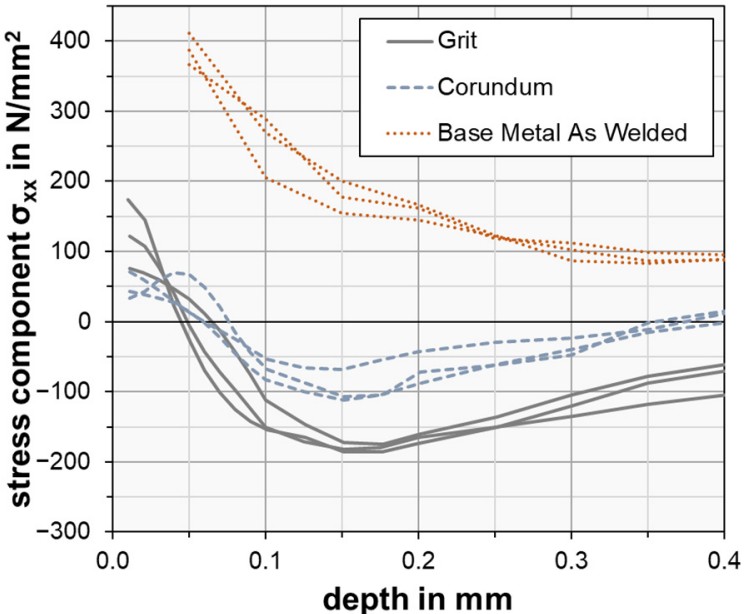

**Figure 8.** The residual stress measurement results of thermally sprayed specimens previously blasted with grit or corundum and non-treated reference specimens in the as-welded state at the weld toe.

*3.2. Fatigue Tests*

Fatigue strength evaluation by statistical analysis with a variable slope according to [40] leads to a characteristic value of the stress range of $\Delta\sigma_{c,var}$ = 153 MPa with a slope of $m_{var}$ = 9.9 for the butt-welded reference specimens ($n$ = 5). The specimens with transverse attachments show a characteristic value of $\Delta\sigma_{c,var}$ = 89 MPa with a slope of $m_{var}$ = 4.0 ($n$ = 9). Blast-cleaned and subsequently thermally sprayed butt weld specimens show a nearly identical characteristic value of $\Delta\sigma_{c,var}$ = 152 MPa ($m_{var}$ = 14.7; $n$ = 8) compared to the non-blasted butt weld specimens. Fractures generally occurred at the weld toe of the specimens. However, with the blast-cleaned and thermally sprayed butt joint specimens, fractures also occurred in the base material area of several specimens ($n$ = 11). Regarding the non-blasted, non-coated specimens, fractures in the base material area did not occur. Blast-cleaned and subsequently thermally sprayed specimens with transverse attachments show a characteristic value of $\Delta\sigma_{c,var}$ = 127 MPa ($m_{var}$ = 6.6; $n$ = 18). For both types of specimens, the kind of surface preparation (grit blasting and blasting with corundum) does not seem to make a difference in the fatigue test results. The corresponding S–N diagrams including the testing parameters and characteristic values of the stress range can be seen in Figure 9.

The different types of crack locations are shown in Figure 10. As shown, for some of the specimens with transverse stiffeners, cracks appeared at both sites of the stiffeners simultaneously.

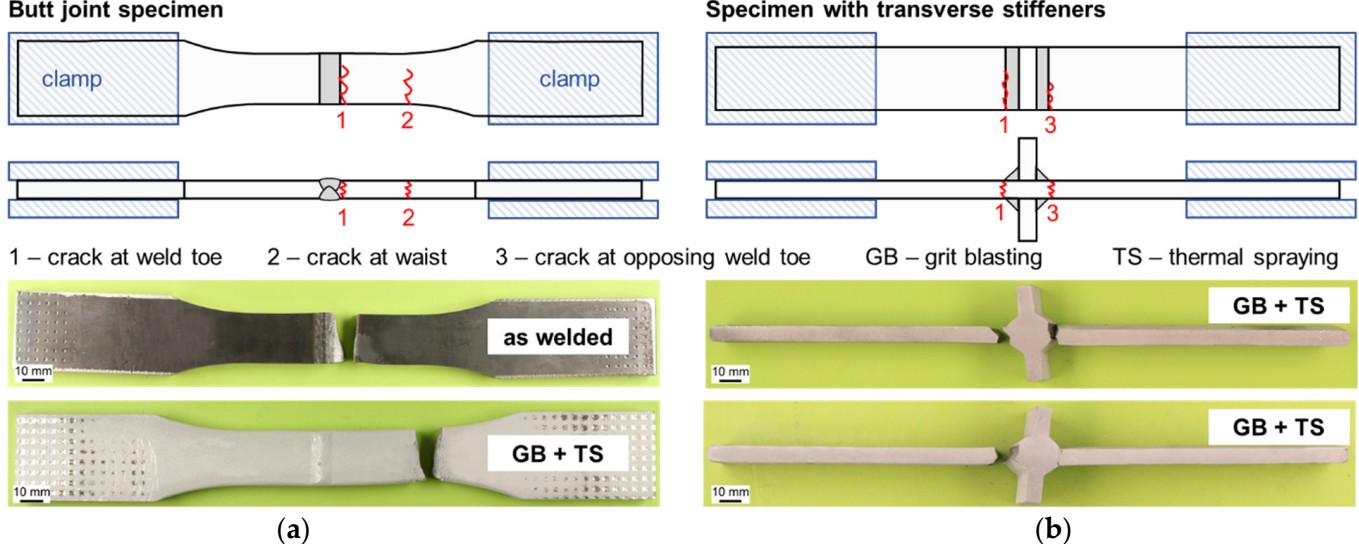

**Figure 9.** S–N Diagrams. (**a**) butt-welded specimens, as welded; (**b**) butt-welded specimens, blast cleaned and thermal spray coated; (**c**) specimens with transverse stiffeners, as welded; (**d**) specimens with transverse stiffeners, blast cleaned, and thermal spray coated.

**Figure 10.** Failure locations. (**a**) Butt joint specimens; (**b**) Specimens with transverse stiffeners.

Concerning the fracture surfaces shown in Figure 11, the fatigue fracture zone with a plain surface orthogonal to load direction and the fast fracture zone with distinct plastic deformation of the material (ductile fracture) can be clearly distinguished. The fatigue fracture zone of substrate and coating is characterized by different types of cracks. While orthogonal cracks with a large opening angle are predominantly observed for the substrate, horizontal cracks are occasionally found in the coating. These mostly occur at the interface or nearby, thus indicating lower adhesion in these areas. Yet, in general the coatings still adhere well to the substrate over most of the observed fracture surfaces. In contrast, orthogonal cracks could not be observed in the coating during the analyses before fatigue testing. Moreover, small particles from the substrate could partly be detected in the coatings (not shown). A single origin of the fatigue fracture could not be found for the specimen shown. Instead, multiple crack fronts originating at multiple points across the weld toe seem to blend together over the width of the specimen. With some of the grit-blasted and thermally sprayed specimens, fatigue cracks occurred at both sides simultaneously or, on the contrary, only at one side of the weld toe. No evidence was found to indicate whether the cracks initiated from the coating or the substrate material. These observations apply to both kinds of surface preparation equally.

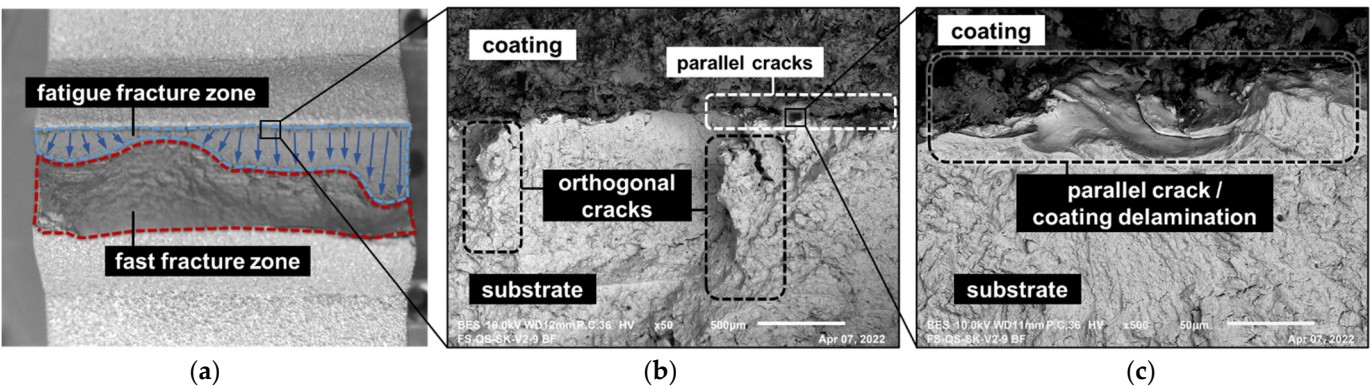

**Figure 11.** The fracture surface of a grit-blasted specimen with transverse stiffeners. (**a**) Overview image showing the fatigue fracture zone and the fast fracture zone; arrows indicate the origin and direction of fatigue crack propagation. (**b**) The BSE image of the interface area showing different orientations of secondary cracks in the fatigue fracture zone. (**c**) The BSE image of the parallel crack at the interface of the coating.

### 3.3. Corrosion Testing

Figure 12a shows the OCP values during the immersion. The steel's OCP started at around $-0.55$ V and stabilized after 3 days of immersion between $-0.65$ V and $-0.67$ V, respectively. During the whole immersion time no further changes in the steel's OCP were observed, except for a decreasing difference between measurements. From the initial values between $-0.80$ V and $-0.86$ V the coating's solo potential declined within 3 to 10 days to less than $-1.0$ V. Subsequently, it increased and seemed to stabilize at around $-0.95$ V by the end of the test duration. The values from the galvanic couples were observed to be in-between these values. The OCP values of the couples started at around $-0.65$ V and declined within 2 to 3 days to values close to $-1.0$ V. At the end of the test duration the combined OCP of the galvanic couples ranged slightly higher than those from the stand-alone coating and were in the region of $-0.9$ V.

The galvanic current density over time is shown in Figure 12b. This was calculated using the surface area of the bare steel specimens. The values of all the couples were initially between 130 mA/m$^2$ and 180 mA/m$^2$. From the outset of the experiment, the values started to rise reaching their maxima after 3 to 5 days. All maxima lay between 400 mA/m$^2$ and 500 mA/m$^2$. After the maximum was reached, the current density declined with the curve being shaped asymptotically. After 20 days of immersion the values of all

the couples were slightly above 100 mA/m$^2$, but still decreasing. At the end of the test all the values were lower than 100 mA/m$^2$.

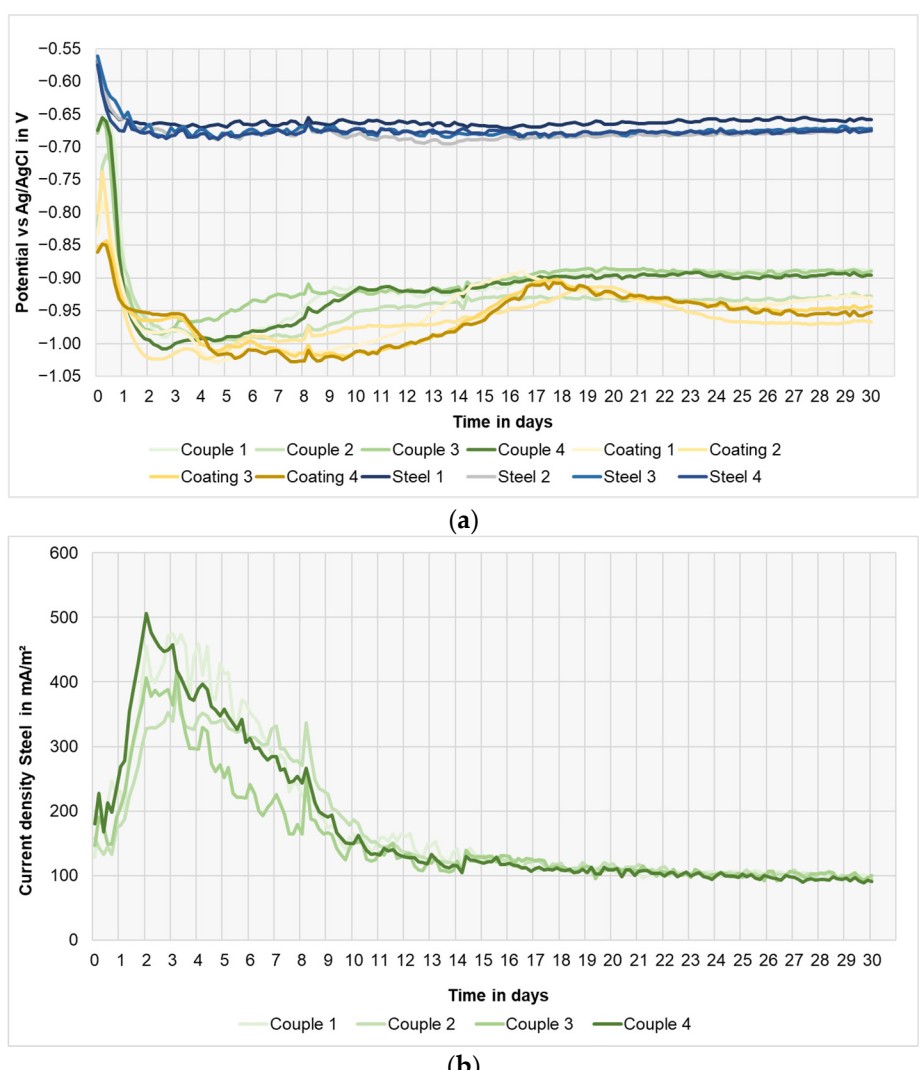

**Figure 12.** (**a**) OCP vs. Ag/AgCl of the single specimens and specimen couples over the test duration; (**b**) Galvanic current density of the thermal spray coated and bare steel specimen couples over exposure time.

Figure 13a presents the difference between a steel specimen that was electrically coupled to a coated specimen on the left side and to stand-alone steel on the right side. The coating cathodically protected the bare steel sufficiently during the test. No signs of corrosion were visible on the surface of any bare steel specimens that were coupled to coated specimens in the wet state directly after being retrieved from exposure. Conversely, a thin layer of corrosion products formed on the earlier-protected bare steel coupled in the dried state. There was no observed effect of the orientation of the steel specimen on the coated plates.

Figure 13b shows the coated specimen that was electrically coupled to the bare steel specimen shown in Figure 13a on the left side. IR Spectroscopy identified the white deposits as aluminum oxide. No optical difference between stand-alone coated specimens and the coated specimens from the galvanic couples after exposure was detected.

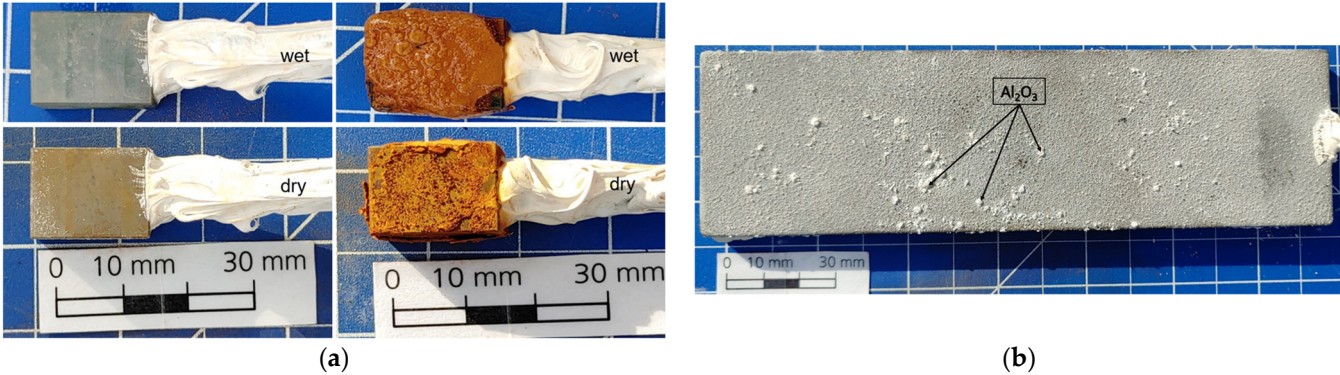

**Figure 13.** (**a**) Bare steel specimens after 30 days of immersion in Baltic sea water; (**b**) a thermal spray coated specimen that was electrically coupled to a bare steel specimen.

The SEM analysis of the steel coupon in the dried state primarily showed iron oxides on the surface. This corroded layer was also optically visible and established during drying after the exposure. In addition, sulfur, magnesium, and sodium were also detected. The SEM images and EDX maps shown in Figure 14 demonstrate an even distribution of the corresponding elements.

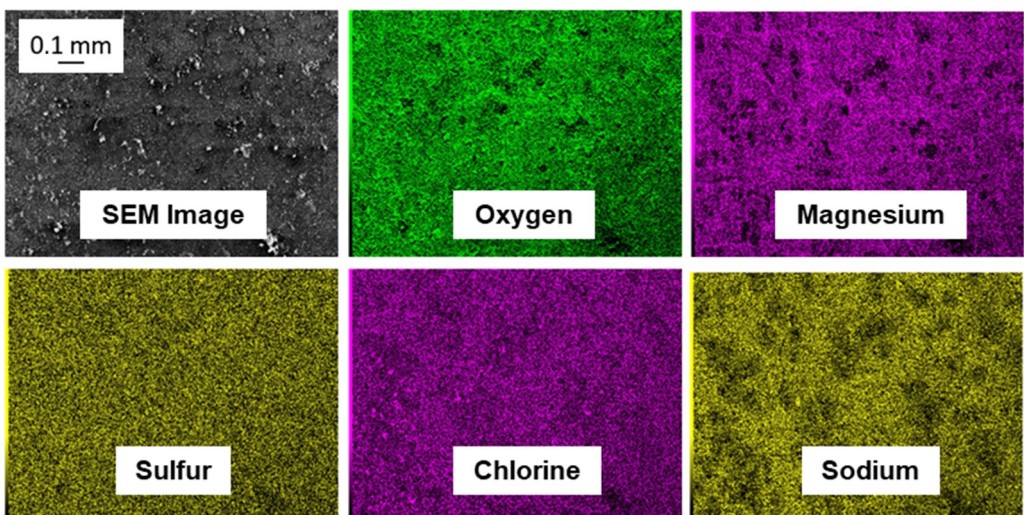

**Figure 14.** The SEM images and EDX maps of a bare steel coupon that was continually galvanically coupled to a coated specimen.

Figure 15a shows the corrosion rates of coated specimens and Figure 15b shows the galvanic couples' corrosion rates during exposure. Initially, the values of the stand-alone coating were lower than those for the galvanic couples. They ranged from 30 μm/a to 50 μm/a. Over the exposure period the corrosion rates decreased to as low as 20 μm/a at the experimental completion. The corrosion rates of the galvanic couples—one could also call them a coating with a 5% defect—were initially higher, ranging from 55 μm/a to 70 μm/a. Instead of directly decreasing after the start of the test, the values rose to a maximum of 80 μm/a. During the 14 days it took the galvanic current density to decrease to 100 mA/m$^2$, the corrosion rates also decreased to around 30 μm/a. At the end of the exposure time the corrosion rates of the galvanic couples reached the same levels as the stand-alone thermal spray coating with the slope still pointing to even lower values.

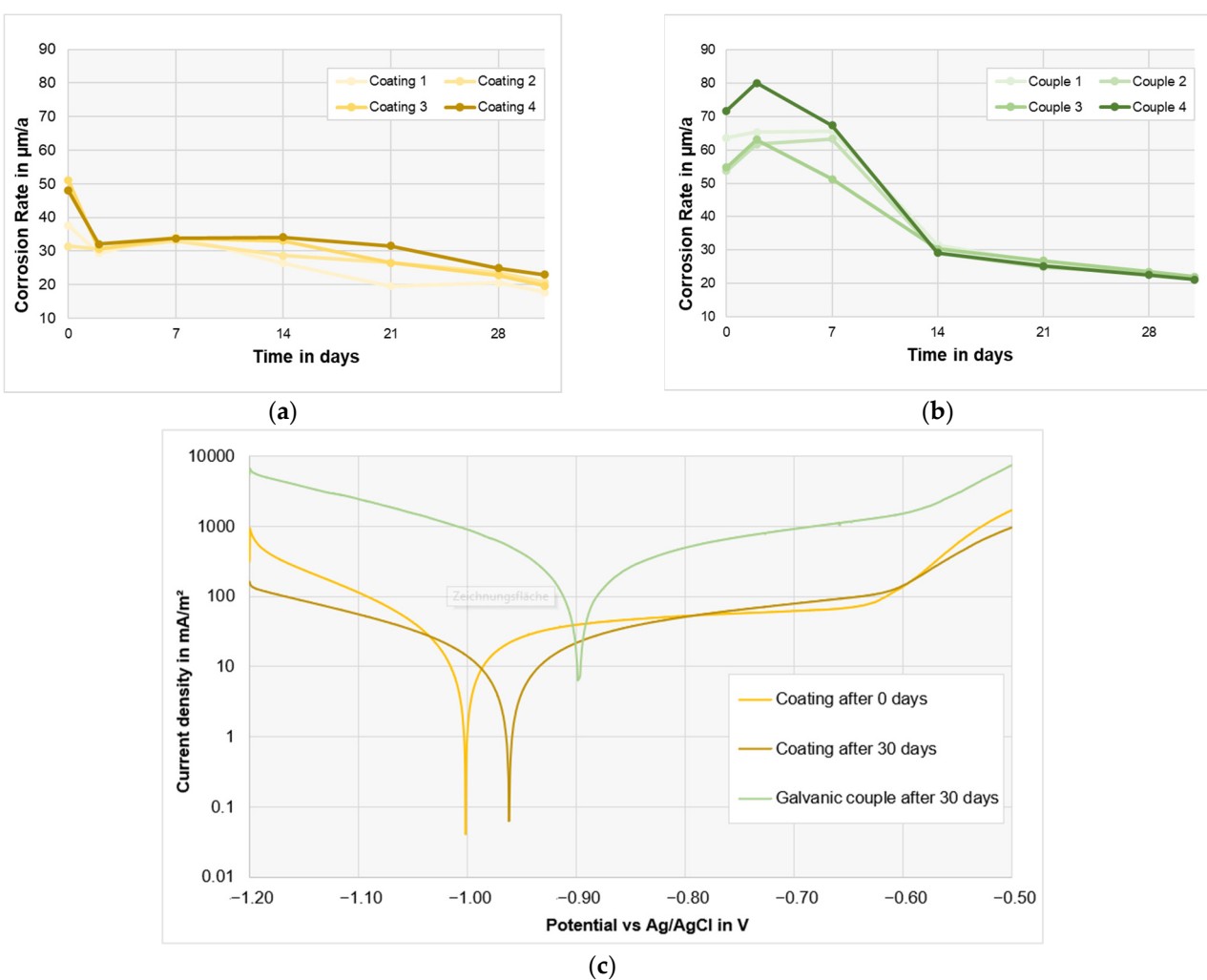

**Figure 15.** (**a**) The corrosion rate of the thermal spray coating; (**b**) The corrosion rate of the galvanic couples; (**c**) Potentiodynamic polarization curves of the coating at the start of the test, and of the coating and a galvanic couple after 30 days.

Potentiodynamic polarization curves of the coating at the start of the test and of the coating and a galvanic couple after 30 days are displayed in Figure 15c.

## 4. Discussion

The fatigue strength of the butt-welded non-blasted specimens is close to base material fatigue strength of $\Delta\sigma_c = 160$ MPa (m = 3) according to EN 1993-1-9 [1]. This surpasses the general recommendation of detail category 90 for butt-welded connections according to the standard by far. This result can be explained by the good execution of the welds with low levels of imperfections, especially regarding angular and axial misalignment, and a general benefit in fatigue strength known for thin-walled specimens [44,45]. The steeper slope and the lower fatigue strength of the as-welded specimens with transversal attachments were expected and are congruent with the regulatory values (detail category 80 for plates with transversal attachments according to [1]). The process of blast cleaning only can lead to different results regarding fatigue strength of the blasted components, depending on the abrasive and blasting parameters used: in prior investigations, blasting with corundum led to a reduction in fatigue strength due to an increased surface roughness and only minor effects on the near surface microstructure, whereas grit blasting led to a significant fatigue strength improvement due to the increased compression of the near surface microstructure, the cold hardening effects, and an increased introduction of compressive stresses [7].

Given that the thermally sprayed specimens show the same fatigue strength characteristics regardless of the blasting process used for the surface preparation, the thermal spraying itself seems to lead to an increase in fatigue strength as well: for the specimens blasted with corundum the thermal spraying seems to compensate for the issues that are normally associated with this kind of surface preparation [7].

Looking at the micrographs, the generally negative effect of increased surface roughness on the fatigue strength of the blast-cleaned specimens might be reduced by the coating material filling out the dents on the substrate surface, which might hinder fatigue crack initiation and growth. This seems to be confirmed by the relative oxide-free coatings, which are instead dominated by clustered porosity. The microstructural analyses and evaluations of the fracture surfaces generally showed a good bonding of the coatings and, in particular, a denser and more uniform coating structure for grit. However, this difference does not seem to have a direct influence on the fatigue strength. Further research is necessary to explain the effect sufficiently. The considerable number of post-weld treated butt-weld specimens with fractures occurring in the base material area shows that the maximum increase in fatigue strength may have been reached for this type of specimen. For these, the reduction of the notch effect of the welds led to other specimen characteristics being more critical, e.g., the tapering of the specimens, which resulted in cracks occurring in the base metal area rather than at the weld toe. For most of the grit-blasted and thermally sprayed specimens, the missing single point of origin of the fatigue cracks indicates a uniform notch effect across the weld toe.

Whilst the Almen intensities of the different blasting processes proved to be on a comparable level, the results of the residual stress measurements confirm the earlier findings already mentioned, which state that blasting structural steel with corundum has more of an abrasive effect and that in contrast grit blasting, with particles larger and more dense than corundum, induces a higher number of compressive stresses into the material and leads to a more distinctive cold work hardening of the near surface areas blasted [5,7]. The more distinct microstructural plastic deformation of the steel substrate surface layer of the grit-blasted specimens compared to the specimens blasted with corundum indicates a stronger cold work hardening effect in the surface area of the grit-blasted specimens and goes along with the residual stress measurement results. The tensile stresses prevailing in the thermal spray coating do not seem to transfer into the blast-cleaned base material nor do they seem to have a negative impact on the fatigue strength of the coated specimens, as the fatigue strength characteristics determined are the same regardless of the kind of blasting process used for surface preparation, or the height of tensile stresses in the specimen coating, respectively. Instead, it might also be assumed, that a certain stress compensation can take place beyond the coating–substrate interface and thus favorable compressive residual stresses could prevail, which is generally in line with the investigation. However, the applied elastic parameters of the residual stress measurements solely apply to one single material and usually cannot include transitions. In addition, parameters for the bulk material were used, although it is known that the Young's moduli of thermally sprayed coatings are in part significantly lower [20,46].

The results of the corrosion test indicate sufficient galvanic protection of areas with defects in Al99% coatings, where bare steel is exposed to sea water with a low salinity. No corrosion products were formed on exposed steel surfaces if they were electrically connected to Al99% coated specimens. During the whole exposure period the OCP values of the stand-alone coating, as well as of the galvanic couples, were below the protection level for bare steel [46].

## 5. Summary

The investigations show that Al99% coatings applied by thermal spraying to welded structural steel specimens of type S355 J2+N lead to a significant improvement in the fatigue strength of the basic structure and that the arc-sprayed coatings show sufficient adhesion even under a fatigue load. For blast-cleaned and coated test specimens with welded

transverse stiffeners, the characteristic value of the stress range was determined with $\Delta\sigma_{c,var} = 127$ MPa at a variable slope of $m_{var} = 6.6$, which is 40 % higher than characteristic value determined for the as-welded reference specimens ($\Delta\sigma_{c,var} = 89$ MPa).

Test specimens with welded butt joints exhibited a very high fatigue strength in the as-welded condition already, which can be attributed to the low level of geometric imperfections of the weld seams and specimens (angular and axial misalignment). Subsequent coating by thermal spraying resulted in a fatigue strength comparable to that in the reference specimens, with $\Delta\sigma_{c,var} = 152$ N/mm$^2$. Crack initiation in the thermally sprayed specimens occurred more in the base material area and less at the weld toe, indicating a reduction of the notch effect of the weld.

The increase in fatigue strength is independent of the blasting intensity and the abrasive used for the surface preparation. However, compressive residual stresses introduced by grit blasting are much higher and extend deeper below the surface compared to blasting with corundum. Also, the cold work hardening effects in the near surface area, which lead to a delay in crack initiation, are more pronounced after blast cleaning with steel grit. Since these differences did not lead to a difference in fatigue strength of the specimens, the thermally sprayed coating itself seems to contribute to fatigue strength improvement as well, e.g., by compensating the generally negative effect of increased surface roughness as well as the lack of sufficient cold work hardening effects associated with the specimens that were blast cleaned with corundum only. This may be attributed to the closure of dents on the substrate surface acting similarly to a reduction in roughness of the specimen surface and/or the stress compensation effects beyond the substrate coating interface and these will be investigated further in future studies.

Concerning the corrosion protection properties, it can be concluded that Al99% coating sufficiently works as a sacrificial anode for exposed steel in aerobic Baltic sea water with a salinity of 0.8% at room temperature. Further tests can be conducted to determine functionality at lower temperatures, in seabed sediment, and for larger defective areas.

Overall, the studies indicate that, in combination, surface preparation by blast cleaning and the application Al99% thermal spray coatings, can sufficiently protect steel structures from corrosion and simultaneously lead to an increase in fatigue strength of welded components. This is especially interesting for, e.g., offshore structures exposed to cyclic loading. The effects examined should be validated in future investigations, for instance in studies that also consider a combination of cyclic loading and corrosive wear.

**Author Contributions:** Conceptualization, K.-M.H. and A.G.; fatigue test specimen manufacturing (supervision of welding and surface preparation), A.G., B.R. and M.H.; thermal spraying (supervision and writing), M.H.; residual stress measurements, base metal surface analysis, and coating analysis (supervision, investigation, writing, and visualization), M.H. and B.R.; base metal properties and fatigue tests (supervision, formal analysis, writing, and visualization), B.R.; fracture surface examination (investigation, writing, and visualization), M.H. and B.R.; corrosion specimen manufacturing and corrosion testing (supervision, investigation, writing, and visualization), J.N. and M.I.; overall supervision, A.G. All authors have read and agreed to the published version of the manuscript.

**Funding:** This research received no external funding.

**Institutional Review Board Statement:** Not applicable.

**Informed Consent Statement:** Not applicable.

**Data Availability Statement:** All data that supports the findings in this study are available to researchers upon reasonable request.

**Acknowledgments:** The authors would like to thank all co-workers involved in the study, namely listed in alphabetical order: R. Arndt, K. Hasche, A. Herhaus, J. Hilbert, F. Knöchelmann, W. Krömmer, and S. Schneider.

**Conflicts of Interest:** The authors declare no conflict of interest.

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
