# Peer review of "Fatigue Strength of Structural Steel-Welded Connections with Arc-Sprayed Aluminum Coatings and Corrosion Behavior of the Corresponding Coatings in Sea Water"

_jmse, doi:10.3390/jmse10111731_

Round 1
Reviewer 1 Report (Previous Reviewer 1)
Dear Authors,
thank you for the professional completion of the manuscript. I believe that it is properly prepared for publication, it contains valuable results. I have no objection to the structure of your work. In addition, I believe that your responses to all reviewers are appropriate. I will support the decision to accept it for publication.
Good luck!
Reviewer
Reviewer 2 Report (Previous Reviewer 3)
The authors have responded to all my questions and have implemented the comments. In my opinion, the manuscript has improved nicely and is worthy of being published. Thanks.
This manuscript is a resubmission of an earlier submission. The following is a list of the peer review reports and author responses from that submission.
Round 1
Reviewer 1 Report
Dear Authors,
Your article titled: “Effect of Thermally Sprayed Aluminum Coating for Corrosion Protection on Fatigue Strength of Structural Steel Weld Seams” is aimed at assessing fatigue strength of two types of MAG welded steel joints for offshore industry. The paper describes the effect of the arc sprayed Al layer on the behavior of joints subjected to cyclic load. I believe that the topic is of great practical importance and work made a positive impression on me. I believe that the subject and content of the article justifies its publication, but some changes should be made. Below I present my remarks and comments in the order they appear in the text.
Affiliations: Please use the full name of the institution (Fraunhofer Institute for Large Structures in Production Engineering IGP, 18059 Rostock, Germany?).
The abstract and keywords do not contain the name of the welding process. As you know, the welding process has a very strong influence on macro- and microstructural morphology and properties of individual zones of welded joints.
The introduction chapter is well prepared, contains the appropriate paragraphs and is clearly written. Please consider adding a paragraph containing a list of factors that affect the reduction of fatigue strength of welded joints used in the offshore industry, e.g. load conditions, material, welding process (presence of defects resulting e.g. from the effect of storage conditions of welding wires on properties of welds ), joint shape, number of passes etc.
Figure 1: in the figure we see 2 photos, and the caption contains the markings a, b and c. This is legible, but in my opinion, the elements shown in photo 1b, c should be described differently.
At the end of the Introduction, please add a clearly formulated purpose of the paper.
Materials and Methods:
To ensure technical correctness, I propose to add the information that the joints with transverse stiffeners are called cruciform joints. Why were these two types of joints chosen? What welding machine was used?
Please add information about the welding process: was the process automatic? Has the heat input been calculated using the coefficient of thermal efficiency?
Terminology: Please use the terms "welding current", "arc voltage".
Has visual examination of the joints (VT) been carried out?
Figure 5 shows the metallographic details: please describe which of the 4 joints variants are shown.
Some markings are too small, e.g. the scale on the axes in figures: 12.
Please check figure 15c for correctness. The terms "Coating start" and "coating end" are not explained in the text.
I suggest replacing the title: Conclusions with Summary. According to the MDPI guidelines, conclusions should be listed in bullet points.
References are well chosen but are not all mentioned in the text.
Please correct the dot insertion point when citing references (after square brackets).
Reviewer 2 Report
The following is my comments for this manuscript.
(1) The abstract should be improved, more detailed data support conclusion should be added.
(2) The whole text is not closely related to the topic and the research purpose of this article.
(3) There are numerous term misusings.
(4) The orrangement of the paper are in a mess, and it is hard to get the point of the author.
(5) The immersion experiment should inculde pre-stressing.
(6) OPC could not be exhibited as average value
(7) The PD curve is not a direct corrosion current data.
(8) What is the purpose to conduct a galvanic experiment?
(9)Where is the most important data for the protection of fatigue behavior?
(10)The paper are more like a data collection than a scientific paper.
Reviewer 3 Report
Review of JMSE-1863743 – Fatigue Strength of Structural Steel Welded Connections with Thermally Sprayed Aluminum Coatings and Corrosion Behavior of Corresponding Coatings in Sea Water
General Comments
This paper presents results about the positive impact of arc-sprayed aluminum coating in protection of steel against protection caused by immersion in water. This was evaluated by measuring the linear polarization resistance. Fatigue strength of the coated samples was also studied. Based on the obtained results, it was concluded that deposition of the aluminum coatings on offshore structures brings about promising advancements in both fatigue strength and corrosion protection aspects.
The reviewer believes that the manuscript is well-written. The importance of the study, the literature, and above all, the novelty are explained clearly. That said, the reviewer believes that this paper requires minor revision.
Title
i. In this study, only one of the thermal spraying means has been used. The reviewer suggests that the term “arc-sprayed” to be used instead of “thermal sprayed” for coatings for accuracy.
Introduction
i. In the literature review section, when explaining about the generated residual stresses, only the mechanism for quenching stresses is explained and there is no mention of residual stresses due to thermal stresses that is caused by the mismatch between the expansion coefficient of the coating and the substrate. The authors are encouraged to add several sentences and references in this regard.
Materials and Methods
i. It is true that the arc spray is the best process for this application due to its low cost of fabrication, low maintenance cost, availability, and being easy to use. However, for the sake of research, it might be a good idea to do the same study by using samples that are fabricated by using plasma spray or HVOF as well. The reason for that is the possibility of having a network interconnected pores when using arc spraying. This might compromise slightly the ideal corrosion protection that is expected.
ii. A torch angle of 10° is considered for deposition of the coatings as listed in Table 1. However, the impact of this angle is not explained. Please provide explanation for the necessity of this angle.
iii. Has preheating of samples been considered in this study? This might impact the generated residual stresses and adhesion of the coating to the substrate.
iv. Has any other material other than aluminum been considered for this purpose? Ni-Cr coatings also provide decent protection against corrosion.
Results
i. The impact of using grit and corundum is nicely shown in Figure 6. It would be very nice to study the impact of grit size on the extent of plastic deformation and generated compressive stresses. This can be done in the next study.
ii. There is a typo in line 301. “b) substrate blasted with blasting grit”
iii. A noticeable variation can be seen in the thickness of the coating in Figure 7(b). Given the large size of the samples shown in Figure 1 and the fact that the coatings are sprayed manually, what mechanisms are used to ensure the uniform thickness of the deposition? The variation can be reduced by increasing the thickness of the coating, but it might affect the adhesion of the coating to the substrate and might lead to delamination.
iv. Throughout of the manuscript, the bonding between the coating and the substrate has been mentioned several times. Mostly, this has been evaluated by looking at the micrographs. It might be a good idea to study this important parameter quantitatively by using ASTM C633 adhesion testing. This can be done in the next study.
v. There is a typo in line 335. “Tensile Compressive stresses in the coating of …”.
vi. Given the compressive stresses are very important for improving the lifetime of the structures that are subjected to fatigue loadings, would it be possible to conduct shot peening prior to grit blasting to reach higher compressive stresses values?
vii. Figure 11 is very informative, and it is taken very professionally. The reviewer suggests adding a high-magnification image of the parallel cracks in the coating if possible.
viii. Figure 14 is not clear, and it is believed that it should be taken again. Please consider increasing the size of the images so that the details can be seen.
Author Response
Dear Reviewer,
thanks for your helpful comments. Please find our response in the attached pdf-file.
Best regards!
Andreas

Round 2
Reviewer 2 Report
Sorry, I must reject this manuscript from my point of view due to its poor quality. The manuscript has no any technic or science improvement after the revision.
Author Response
- Unfortunately, we cannot provide a substantive response to this reviewer's comment. From the authors' point of view, it is remarkable and difficult to understand that after the extensive consideration of the reviewer's comments from the first review (see below), a worse overall rating was given compared to the first round of reviewing from the same reviewer. This would mean that the revision of the paper according to the reviewer comments made the article even worse.
- We therefore ask for specific comments and advice in order to meet the requirements of the reviewer.
- We would also like to point out that the other two reviewers do not share this reviewer's general opinions regarding the basic technical and scientific approach, investigations and paper preparation.
